# Underwater Multispectral Laser Serial Imager for Spectral Differentiation of Macroalgal and Coral Substrates

**Matthieu Huot** [1,*]**, Fraser Dalgleish** [2]**, Eric Rehm** [3]**, Michel Piché** [4] **and Philippe Archambault** [1]

1   Département de Biologie, Takuvik, Québec-Océan, Université Laval, Quebec, QC G1V 0A6, Canada; philippe.archambault@bio.ulaval.ca
2   BeamSea Associates, Loxahatchee, FL 12823, USA; fdalgleish@beam-sea.com
3   Sea-Bird Scientific, Bellevue, WA 98005, USA; eric.rehm@gmail.com
4   Département de Physique, de Génie Physique et D'optique, Université Laval, Quebec, QC G1V 0A6, Canada; michel.piche@phy.ulaval.ca
*   Correspondence: matthieu.huot.1@ulaval.ca; Tel.: +1-418-750-1262

**Abstract:** The advancement of innovative underwater remote sensing detection and imaging methods, such as continuous wave laser line scan or pulsed laser (i.e., LiDAR—Light Detection and Ranging) imaging approaches can provide novel solutions for studying biological substrates and manmade objects/surfaces often encountered in underwater coastal environments. Such instruments can be used shipboard or coupled with proven and available deployment platforms as AUVs (Autonomous Underwater Vehicles). With the right planning, large areas can be surveyed, and more extreme and difficult-to-reach environments can be studied. A prime example, and representing a certain navigational challenge, is the under ice in the Arctic/Antarctic or winter/polar environments or deep underwater survey. Among many marine biological substrates, numerous species of macroalgae can be found worldwide in shallow down to 70+ m (clear water) coastal habitats and are essential ecosystem service providers through the habitat they provide for other species, the potential food resource value, and carbon sink they represent. Similarly, corals also provide important ecosystem services through their structure and diversity, are found to harbor increased local diversity, and are equally valid targets as "keystone" species. Hence, we expand current underwater remote sensing methods to combine macroalgal and coral surveys via the development of a multispectral laser serial imager designed for classification via spectral response. By using multiple continuous wave laser wavelength sources to scan and illuminate recreated benthic environments composed of macroalgae and coral, we show how elastic (i.e., reflectance) and inelastic (i.e., fluorescence) spectral responses can potentially be used to differentiate algal color groups and certain coral genus. Experimentally, three laser diodes (450 nm, 490 nm, 520 nm) are sequentially used in conjunction with up to 5 emission filters (450 nm, 490 nm, 520 nm, 580 nm, 685 nm) to acquire images generated by laser line scan pattern via high-speed galvanometric mirrors. Placed directly adjacent to a large saltwater imaging tank fitted with optical viewports, the optical system records target substrate spectral response using a photomultiplier preceded by a filter and is synchronously digitized to the scan rate by a high sample rate Analog-to-Digital Converter (ADC). Acquired images are normalized to correct for imager optical effects allowing for fluorescence intensity-based pixel segmentation via intensity thresholding. Overall, the multispectral laser serial imaging technique shows that the resulting high resolution data can be used for detection and classification of benthic substrates by their spectral response. These methods highlight a path towards eventual pixel-wise spectral response analysis for spectral differentiation.

**Keywords:** multispectral; laser; imaging; fluorescence; automated; discrimination; macroalgae; coral

## 1. Introduction

Shallow coastal underwater biological surveys make for an essential part of active ecological and environmental research. Numerous in situ methods for collecting relevant

biological, physical, and phenomenological environmental data exist (e.g., photo id survey, species % cover, bottom substrata id, behavioral video) and are a vital aspect of classical research methods. With technological advancement, some of these methods have evolved to distance-based, or more specifically Remote Sensing (RS) measurements such as satellites, airborne, and Aerial Unmanned Vehicles (AUVs). These methods are useful in partly circumventing often complicated field work (e.g., diving surveys), at the expense of intricacies related to the use of RS instrumentation. For example, high-resolution satellite imagery allows characterization of benthic biological substrates such as coral [1–3], macroalgae [4], eelgrass [5,6], and others. These remain somewhat limited in their resolution to a few, if not many meters, and result in a loss of detail and spectral information via water column scattering and absorption effects. To help mitigate these effects, imaging can be done while reducing distance to surface and can be performed on days with better imaging conditions (e.g., calm water surface). Airborne (i.e., plane) surveys are also presently done for studies of macroalgae [7,8], eelgrass [9] distribution, coastal habitats [10] and aerial AUV surveys performed on macroalgae using hyperspectral imaging sensors [11,12] which usually provide better resolution than satellites. These are considered somewhat still limited in resolution required for underwater classification and identification at a smaller scale than a stand (e.g., kelp stand, eelgrass patch).

To improve resolution for underwater RS and bring further light to the sub-surface biological environment, instrument adaptations must be made. Passive illumination from the sun, which most satellites use to measure reflectance and spectral response, causes a remote sensing instrument to lose much of its resolution when probing in deeper than shallow water. Besides absorption and scattering by water molecules, particles in the water column can be an important factor in scattering of the downwelling (i.e., illumination) and upwelling (i.e., reflectance, fluorescence) light, also leading to a reduction of resolution and contrast. This type of illumination underperforms for underwater imaging for these reasons. Comparatively, active illumination systems can vary in form and use, such as a broad light source in an underwater photo/video imager [13], up to a focusable laser, as in certain satellites (e.g., ICESat-2 or planes equipped with LiDAR (Light Detection and Ranging) sensors. While both may be used to image remotely following illumination and even reaction (e.g., fluorescence), the latter can generate a spectral response in the observed substrate over a small footprint area (i.e., cm to km, depending on application), but also provide range (e.g., sea-level/shallow depth bathymetry [14].

The advantages of direct illumination of photo responsive underwater biological organisms by laser source have seen much evolution over the past 30 years. Many of these efforts have been geared towards deployment by divers or onboard semi-autonomous underwater instruments [13,15–19], but efforts are now mostly on integrating these imaging technologies into underwater AUVs or appropriately fitted research vessels. Naturally, an efficient way to generate spectral response for biological substrate classification is by bringing laser imaging to underwater RS via underwater AUV integration (i.e., Continuous Wave (CW) Laser Line Scan (LLS), or CW-LLS; serial LiDAR or pulsed LLS). Laser imaging systems like these are especially well suited for working in different lighting environments, such as in the dark, where passive illumination would not work, but also in less than optically clear waters (e.g., serial LiDAR). Reducing the light source beam angle by using a tightly focused laser beam reduces scattering volume in water, resulting in better image contrast [20]. Such detection methods can also be applied in more extreme environments, such as under ice-covered marine environments. Advantages of using an AUV also include the possibility of simultaneous oceanographic measurements such as temperature, salinity, water column chl-a, nutrients, irradiance, etc.) and using these for radiometric correction.

Response to light in a substrate can be observed through water but possibility of its detection is wavelength dependent (notwithstanding particle scattering effects), due to absorption processes, especially towards the infrared where photons become exponentially absorbed by water molecules [21]. Many aquatic photosynthetic/photo responsive organisms such as phytoplankton, diatoms, macroalgae, and corals have evolved to using or

reacting to available light within the PAR range (i.e., 350/400–700/800 nm) for photosynthesis. This process allows their study by observing fluorescence as a by-product. Reflectance measurements in underwater substrates or organisms can also be an important tool for target discrimination. Several lab studies have shown the potential for fluorescence detection and possibly identification in color class of macroalgae [17,22,23]. Although thousands of species of macroalgae are found worldwide, all belong to one of three possible color classes (Rhodophyta: red, Chlorophyta: green, Phaeophyta: brown), each type with their own characteristic photopigment assemblages. The latter are responsible for color class-specific wavelength-dependent spectral response to light (i.e., reflectance, fluorescence, absorption), with some variations between species and depending on health status. Corals may also vary by their reflectance and fluorescence depending on species with/without symbiotic microalgae and/or structural elements and fluorescent proteins [24–26]. There is therefore much potential to benefit from these processes in devising ways to detect and classify them.

To improve upon current automated underwater imaging techniques for classification of coastal benthic flora and sessile fauna, our work on underwater multispectral laser serial imaging demonstrates the potential for discriminating between different macroalgal color types and coral under different imaging scenarios. Further, we emphasize the photobiology of macroalgae and coral in selecting suitable instrumentation for generating and recording their spectral response (i.e., reflectance, fluorescence), while attempting to keep a simple design. Image processing techniques related to the methods used are discussed in detail, including detector illumination falloff correction and pixel segmentation processes that can lead to classification. Results on reflectance and fluorescence measurements, including the possibility of evaluating practical fluorescence efficiency, are discussed while proposing possible improvements to the imager for better substrate discrimination.

## 2. Materials and Methods

### 2.1. Laser Imaging Setup

Laser imaging systems typically consist of an emitter and detector assembly working in synchrony for signal generation and acquisition. Various options exist in the configuration of the elements within such a system, such as CW-LLS or using pulsed-gated (PG) as for LiDAR applications. While galvanometric mirrors were our choice for creating a line scanning and reproducing a moving platform (i.e., 2-axis scanning), a rotating prism could also have been used, as well as a Micro-Electro-Mechanical System (MEMS) optical beam steering. However, the prism limits scanning to one dimension without an added linear displacement of the scanning assembly but may be appropriate once the system is mounted to an imaging platform capable of directional movement. The MEMS can be more fragile in field deployed instruments, but still a valuable option. Choice of detector can also depend on the intended application, where PMTs may offer the most signal-to- noise ratio in a dark environment but can be replaced by Silicon Photo Multipliers (SiPMs) in situations where imaging is done in the daytime and sensors may be subject to a very wide range of illumination. The deployment platform also imposes certain restrictions, such as payload volume and overall power consumption. In the context of applying this type of instrument on a limited space and low power AUV, we chose to experiment with laser diodes as they fit these requirements, are affordable and much less restricting for wavelength selection range. A PMT with high red wavelength sensitivity and relatively equal Quantum Efficiency in the PAR 400–700 nm range (approx. 10 to 11%) was our choice to maximize spectral response detection in a controlled setting. Narrow bandpass filters are typically necessary in isolating specific inelastic (i.e., fluorescence) responses from elastic (i.e., reflectance) response and ambient underwater light, as they can all be present to a varying degree during each sampling event. A schematic is shown in Figure 1. Further specifics on the multispectral laser sources, scanning, and detection assemblies as well as data acquisition procedures are explained in detail as Supplementary Materials (see Supplementary Materials section Additional methods M1).

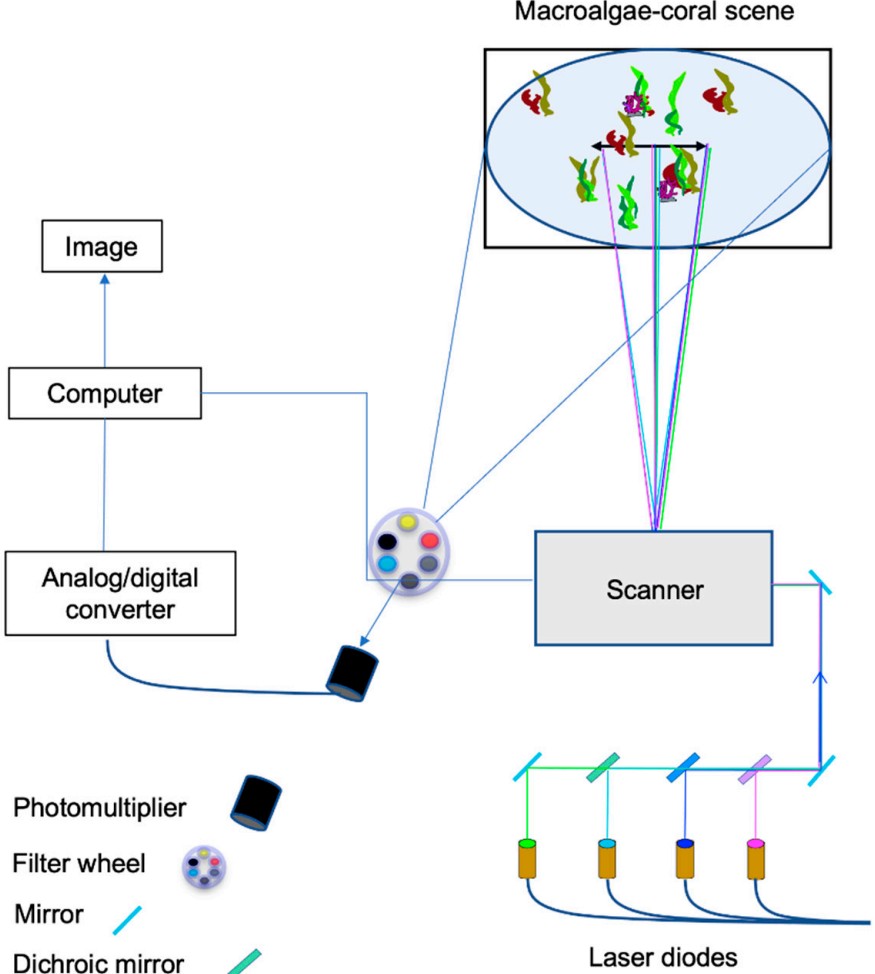

**Figure 1.** Multispectral laser serial imager experimental setup, showing aligned laser diode array, 2-axis galvanometric scanner, underwater imaging scene behind fused borosilicate optical port, PMT, and filter wheel assembly, linked to an ADC unit for signal acquisition and digitization to a computer via a custom GUI.

### 2.2. Imaging Environment

Experimental Saltwater Tank/Benthic Scene Recreation

The saltwater experimental tank in which we performed underwater imaging tests has dimensions roughly 1.5 mW × 7.0 mL × 1.5 mH. The tank is divided into three compartments, with one bulkhead at each 1/4 distance from the tank extremities. Each bulkhead is fitted with two fused silica optical ports (approx. 150 mm diameter), these being placed at 0.5 m from the tank bottom, distanced from each other by 0.6 m and themselves centered on the bulkhead at their mid-distance. The central compartment is fitted with 4 vertically aligned inlet nozzles per corner and one intake near the tank bottom, with the possibility of controlling water flow volume and direction at each nozzle. The 4 outlets are connected to a $\frac{3}{4}$ HP pump which takes the tank water through a filtration system consisting of a UV filter (Smart HO UV sterilizer, 150 Watt), a 5-gallon activated carbon filter and a 1-micron bag filter operating at 20–25 psi pressure.

Live macroalgae were obtained from the low tide to shallow subtidal zone of the Florida Atlantic coast in proximity to Harbor Branch Oceanographic Institute (Fort Pierce, FL, USA), Wabasso, Golden Sands State Park and South Beach, Vero Beach, Florida. Specimen selection was based on visual appearance, where color, turgidity, lack of necrosis and no/low epiphytic growth were the main criteria. Following collection, samples were brought to our imaging facility, gently washed to remove any sediment and epiphytic

growth, and placed into the experimental saltwater tank (Figure 2a). Macroalgae belonging to the red (Rhodophyta), green (Chlorophyta), and brown (Phaeophyta) algae classification groups were selected (6 total—Appendix A Table A1) to generate a series of spectral response images based on their known characteristic fluorescence and absorption responses. As characteristic photopigment assemblages are specific to these three color groups, it is expected that these spectral responses can be generalized to other species of the same three groups and can be applicable worldwide.

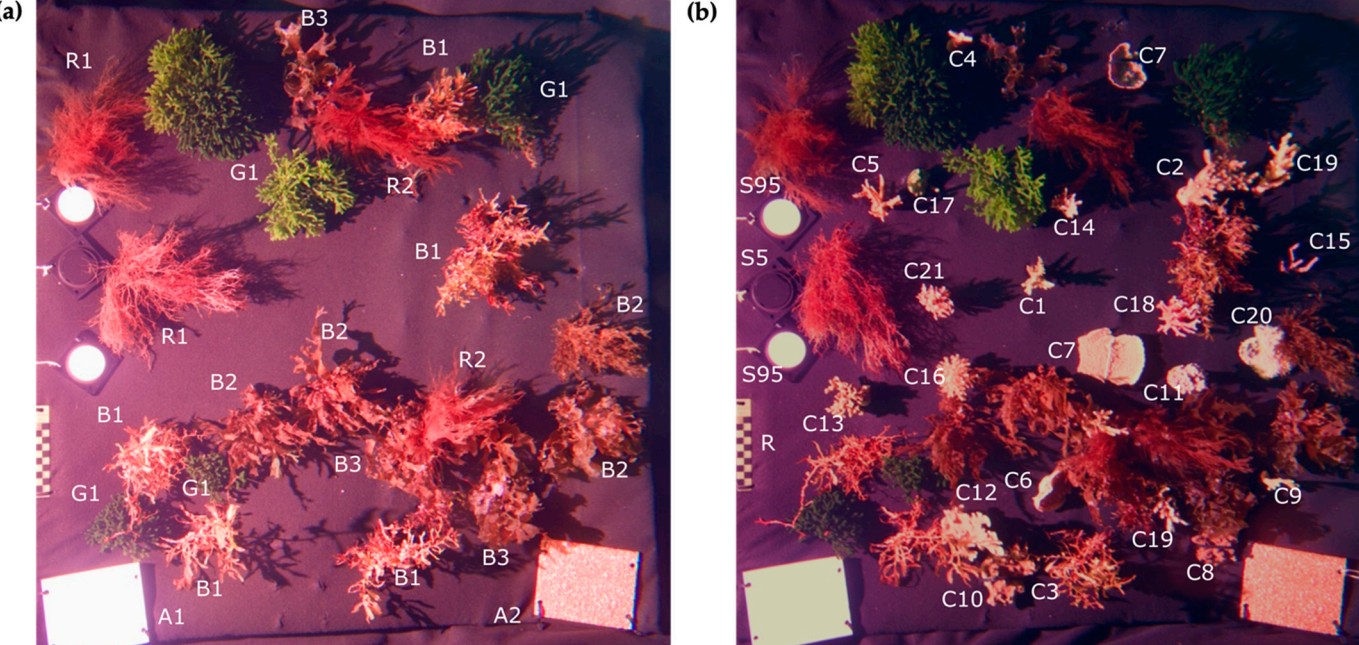

**Figure 2.** Macroalgae and coral imaging substrates arrangement prior to imaging tests where (**a**) macroalgae-only scene to left and (**b**) mixed macroalgae-coral scene to right. Shown are Green: (G1) *Codium* sp.; Brown: (B1) *Sargassum* sp.; (B2) *Dictyota* sp.; (B3) *Padina* sp.; Red: (R1) *Grateloupia* sp.; (R2) *Halymenia* sp.; Coral: (C1) *Acropora (A) austrera*; (C2) *A. cyatherea*; (C3) *A. nasuta*; (C4) *A. nobilis*; (C5) *Acropora valida*; (C6) *Echinopora lamellosa*; (C7) *Montipora (M) capricornis*; (C8) *M. confusa*; (C9) *M. digitata*; (C10) *M. spongodes*; (C11) *Nephthea* sp.; (C12) *Pavona (P) decussatus*; (C13) *P. frondifera*; (C14) *Pinnigorgia flava*; (C15) *Plexaura flexuosa*; (C16) *Pocilliopora damicornis*; (C17) *Psammocora stellata*; (C18) *Seriatopora hystrix*; (C19) *Stylophora pistillata*; (C20) *Xenia umbellata*; (C21) *A. nana*; (A1) Artificial surface 1; (A2) Artificial surface 2; (R) Fluorescence scale; (S95) 95% Spectralon reference; (S5) 5% Spectralon reference.

In addition, multiple live coral specimens of different species (20 total—Appendix A Table A1), obtained through a local supplier of sustainably cloned coral fragments (see ORA production and husbandry practices), were as well incorporated onto the benthic scene (Figure 2b). To provide stable imaging conditions, specimens were affixed to a non-fluorescing semi-flexible Vexar® plastic mesh structure measuring approximately 1.0 m × 1.0 m, which was initially covered by a thick black felt material to provide a uniform, non-reflective/fluorescing black background to the live substrates. For both macroalgae and coral, water flow was adjusted to generate a laminar-type current (e.g., 20–30 cm s$^{-1}$) over the macroalgae and coral substrates, leading to slight movement in their part but not in excess, which could possibly lead to their stress or detachment.

### 2.3. Imager Spectral Characteristics

#### 2.3.1. Contrast-Related Image Quality Metrics

Multispectral image quality was evaluated via Contrast Ratio (CR) and Contrast Signal-to-Noise-Ratio (CSNR) metrics. Specifically, 20 × 20 pixel matrices were extracted from

illumination falloff corrected images' targets corresponding to 99.9% and 5% reflectance Spectralon™ pucks and nearby empty background from elastic images (i.e., 490 nm excitation and 490 nm emission) Spectralon™. The following equation was used to calculate the Contrast Ratio:

$$\text{Contrast ratio} = \frac{\text{mean (white)}}{\text{mean (black)}} \tag{1}$$

where mean (white) and mean (black) represent the mean of Spectralon™ pixels and the mean of nearby background pixels, respectively.

The following equation was used to calculate the Contrast Signal-to-Noise-Ratio:

$$\text{CSNR} = \frac{\text{mean (white)} - \text{mean (black)}}{\sqrt{\text{stdev (white)} + \text{stdev (black)}}} \tag{2}$$

where stdev (white) and stdev (black) represent the standard deviation of Spectralon™ pixels and the standard deviation of nearby background pixels, respectively.

### 2.3.2. Fluorescence-Related Imaging System Variance Measure

To characterize fluorescence response imaging reproducibility, the macroalgal and coral benthic scene was imaged a total of five times at each excitation–emission combination explained in detail in Section 2.2. These repeated measurements allow for a pixel-wise intensity variance evaluation between images and can easily be visualized as a pixel-wise intensity variance map between replicate images.

### 2.3.3. Reflectance and Practical Fluorescence Efficiency Estimation

In an ideal scenario, photon flux and irradiance on the pixel can be precisely determined by constant monitoring of the outgoing laser power output during scanning. While this was not feasible for this experiment, a means to estimate practical fluorescence efficiency from these measurements via photon budget can be conducted by the method described in Supplementary Materials Additional Methods M1. However, reflectance standards were included in the imaging scenes, and allow us to infer pixel reflectance and fluorescence. By imaging these calibrated reflectance standards (i.e., Spectralon™ pucks) in conjunction with the macroalgae and coral specimens, reflectance standard (i.e., 5% and 95%) reflection intensity can be used to evaluate reflectance and fluorescence in other substrates.

The efficiency, or yield, is considered relative, as a typical underwater LLS or LiDAR imaging system will only have access to the elastic and inelastic backscatter from a $2\pi$ sr solid angle since seeing past/under an object is not practical or easily feasible. How much of the inelastic forward scatter (i.e., behind the imaged substrate) is generated during a scan event remains unclear and dependent on what happens inside the structure (if transmission is possible) but much is likely lost through re-absorption within substrate structure (e.g., macroalgae blade, coral structure—depending on shape). Fluorescence emitted at or near 685 nm by chlorophyll-a is readily re-absorbed by adjacent photosynthetic structures if conditions allow it, for example, from within an algal structure.

In a laser line scan-generated image, the number of photons emitted, either via reflectance or fluorescence, by the surface can be estimated by calculating pixel intensity value change from the outgoing light reaching the target and, in this case, the wavelength-dependent transmitted elastic backscattered values of the known reflectance standard pucks. This allows the estimation of average reflectance in other substrates, or the received elastic photons.

Similarly, the number of inelastic photons emitted from a surface can be estimated by considering the observed inelastic signal intensity value, while considering for the emission wavelength since photons originating at different wavelengths will have different energy levels for a same number of photons. In a situation where the number of elastic photons is known (i.e., via power monitoring) to correspond to an observed signal intensity, the

corresponding number of observed photons at another wavelength, such as at 685 nm, can be estimated.

In both situations, whether the number of incident photons is known, or the pixel intensity is used as a proxy of the number of wavelength-dependent photons, beam attenuation coefficient, c, (otherwise a + b: absorption + scattering) must be accounted for the outgoing laser beam to correct for wavelength-specific water column absorption and scattering from the source to the target scene. Moreover, Kd (z, λ), the downwelling diffuse attenuation function (i.e., decrease of the downwelling irradiance, Ed (z, λ), by depth m$^{-1}$) must also be applied to the observed elastic and inelastic signal values to more precisely estimate the number of elastic and inelastic photons originating from the illuminated substrate. In this study, water column attributes correspond closely to Jerlov 1A waters following particulate filtration and UV sterilization system activation in preparation for imaging (i.e., values taken from [27]: 490 nm ex; aw = 0.0196, bsw = 0.0031, Kd = 0.0212). Additionally, care must be taken to consider the spherical spreading factor, $1/r^2$, where r = distance to target (assuming isotropic response). Wavelength-dependent PMT Quantum Efficiency (QE) can also be compensated for before comparing elastic to inelastic photons for a more thorough estimation of practical fluorescence.

In theory, the practical fluorescence yield, or efficiency, can be calculated as follows (see [28] for a detailed explanation):

$$\text{Practical Fluorescence Efficiency } (\Phi_{\text{P fl}}) = \frac{\text{Nb inelastic photons emitted by surface}}{\text{Nb elastic photons emitted} - \text{Nb inelastic photons emitted}} \qquad (3)$$

Estimating the number of observed elastic and inelastic photons can notably be done by working in photon units, from the initial number of photons emanating from the imaging source. Another method is by working in relative photon units, using a calibrated sensor (e.g., considering for wavelength-specific cathode radiant sensitivity). Pixel data preparation done for calculating the practical fluorescence efficiency are listed in order: (1) the image minimum and maximum pixel values, as well as zero offset are calculated using the 5% and 99% reflectance standards. This is initially useful in calculating the background material reflectance, for comparison to other substrates after normalizing their pixel values to image minimum and maximum intensity range; (2) compensation for background reflectance variation due to illumination falloff in an area of the image is performed by first choosing a subset of pixels near the substrate to be evaluated (e.g., coral or macroalgae). Since the background is of the same material and reflectance and fluorescence properties remain the same throughout the image, an "illumination correction factor" is calculated from subset pixels to correct for illumination discrepancies in the background, as well as for substrates of interest; (3) the number of elastic photons emitted from the surface can be approximated by comparing pixel values on a given area to values calculated via the reflectance standards, considering for beam attenuation and diffuse attenuation K:

$$\text{Nb elastic photons emitted}_\lambda = \text{Nb elastic photons}_{\text{at source}} \, c \, K_{z \, (\lambda \text{ elastic})} \, \text{Substrate reflectivity} \qquad (4)$$

Following this step, (4) an estimation of the number of inelastic photons emitted (e.g., 685 nm) by the illuminated (e.g., at 490 nm excitation) substrate, or in other words, the Practical Fluorescence Efficiency, $\Phi_{\text{P fl}}$, can be obtained by bringing the results of the previous 3 equations together into the following equation:

$$\text{Nb inelastic photons emitted }_{\lambda \text{ inelastic}} = \frac{\text{Nb elastic photons emitted}_{\lambda \text{ elastic}} \, \frac{\text{Inelastic image pixel value}}{a_\lambda + b_\lambda}}{\frac{\text{Elastic image pixel value}}{a_\lambda + b_\lambda}} \qquad (5)$$

or

Practical Fluorescence Efficiency $(\Phi_{P\,fl})$

$$= \frac{\left(\dfrac{\frac{\text{Macroalgae inelastic pixel intensity}}{K_{z_{685}}}}{\text{Spherical spreading factor}}\,685\text{ nm}\right)}{\left(\dfrac{\frac{\text{Maximum pixel intensity (i.e., 100\% reflectance standards)}}{K_{z_{490}}}}{\text{Spherical spreading factor}}\,490\text{ nm}\right)\left(\dfrac{\frac{\text{Target substrate elastic pixel intensity}}{K_{z_{490}}}}{\text{Spherical spreading factor}}\,490\text{ nm}\right)} \qquad (6)$$

$$\text{Practical Fluorescence Efficiency }(\Phi_{P\,fl}) = \frac{\text{Relative nb photons fluoresced by substrate}}{\text{Relative nb photons absorbed by substrate}} \qquad (7)$$

### 2.4. Image Processing

Per imaging scenario (i.e., macroalgae only, macroalgae + coral, macroalgae + coral + manmade target), 12 images were acquired by laser scanning and simultaneously recording spectral response at different excitation and emission wavelengths. Combinations considered for multispectral analysis were: (1) 450 nm excitation + 450 nm, 488 nm, 520 nm, 580 nm, 685 nm emission; (2) 488 nm excitation + 488 nm, 520 nm, 580 nm, 685 nm emission, and 3) 520 nm excitation + 520 nm, 580 nm, 685 nm emission.). Each image consists of the same number of pixels (i.e., 1000 by 1000, 1,000,000 total). Per imaging scenario, these images can be represented as part of a multispectral stack 12 layers thick for which each pixel location on the stack has 12 levels of information (Figure 3). As a way of measuring imaging system reliability, 5 replicate scans were performed during the same imaging session while iterating through each of the excitation–emission pairings (As a way of measuring imaging system reliability, 5 replicate scans were performed during the same imaging session while iterating through each of the excitation–emission pairings).

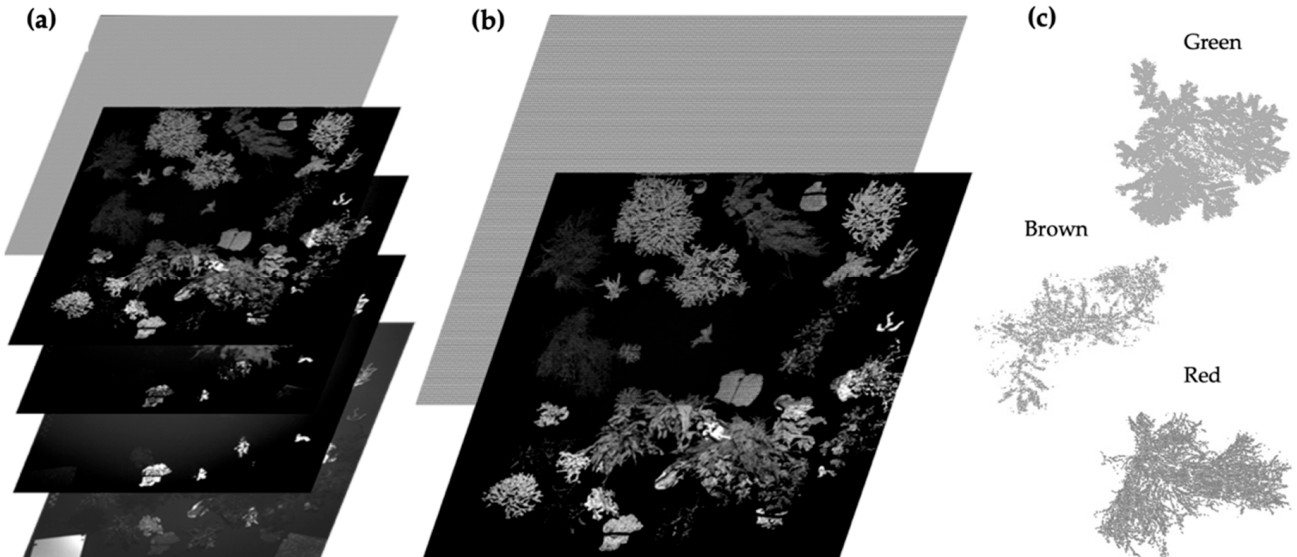

**Figure 3.** Representation of mixed species macroalgae and coral multispectral response stacked data set arrangement (**a**) bottom to top: 490 nm excitation, with 490 nm, 520 nm, 580 nm, and 685 nm emission + pixel grid; (**b**) detailed 685 nm emission from 490 nm excitation + pixel grid; (**c**) isolated pixels belonging to three types of macroalgae after segmentation (i.e., segmentation of fluorescent pixels via intensity value thresholding).

The acquired data layers initially underwent correction steps for detector optical effects. Mainly, images appeared to exhibit a roll-off of luminance or vignetting, where fewer light rays are reaching the sensor the further from the FOV center they originate on the scan field. While this was not accounted for in this prototype multispectral imager, such effects can be minimized by using a matching lens. The drop-off in light intensity towards

the edges of the images can also be somewhat corrected by removing the background illumination by a technique involving imaging a homogeneous target which is possible by using the same optical imaging system [29]. As we imaged a checkerboard pattern reference background for illumination correction and not a uniform background (which would have been easiest), fitting a curve matching the illumination peak of the reference background (and hence each white square of the checkerboard) for each excitation wavelength was necessary to capture the illumination intensity curve. To verify image intensity correction, background values (i.e., between algal, coral, and artificial substrates) were compared and therefore images deemed corrected when little or no illumination gradient was present in the background. Light falloff could also be minimized by restraining image dimensions to the 65% inner pixels, where light falloff could be more easily compensated for. Importantly, no other radiometric corrections were made to the acquired images, keeping them intact for pixel-wise spectral response comparison and analysis.

The process of image normalization (Figure 4) was followed by intensity thresholding to identify pixels showing fluorescent response. Thresholding for macroalgae segmentation was conducted by using spectral response channels showing the strongest response, from excitation at 490 nm and emission at 580 nm (red macroalgae) and 685 nm (green macroalgae, brown macroalgae). Coral pixel identification via programmatic thresholding was performed by using 450 nm and 490 nm excitation, and 490 nm, 520 nm, 580 nm, and 685 nm emission responses, where many channels showed similar but slightly different responses, hence slightly different pixel segmentation between channels. This can eventually be a cause for classification error, as spectral response channels do not overlap perfectly between excitation wavelengths. In an eventual spectral response classification scenario, manual segmentation of the pixel identities such as algal color class and specimen id, coral type/species group could be necessary to validate classification algorithms. Illumination-normalized images can thereafter be used for unsupervised and supervised classification.

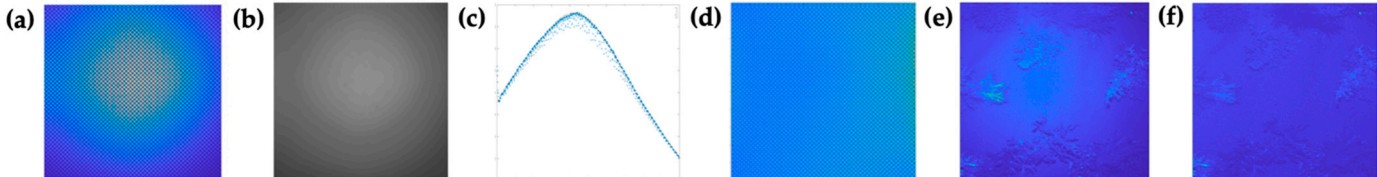

**Figure 4.** Illumination falloff correction procedure for 1000 by 1000 pixel images (except (**e**,**f**), which are subset of 650 by 650 pixels, centered on image), showing (**a**) background checkerboard; (**b**) intensity averaged background (i.e., Gaussian filter—10 pixel radius); (**c**) curve-fitting pixel intensity profile at center of illumination; (**d**) original checkerboard background corrected for illumination falloff; (**e**) uncorrected raw image with noticeable light intensity falloff; (**f**) illumination falloff corrected image where background is mostly uniformized in its intensity.

## 3. Results

### 3.1. Image Normalization

The image normalization process was efficient in noticeably reducing illumination falloff effect in spectral response images. However, it must be mentioned that the checkerboard background was only imaged in the absence of filters. Hence, the illumination falloff correction shows better results in the reflectance images, that is, when no filters were in use, than for other filters (Figures 5–7). This effect is more noticeable due to an open-diameter reduction by the filter "border ring", and under-correction is due to this on image borders but especially the corners where the effect is more noticeable. Figures 5–7 may also have been altered for their intensity or contrast and de-speckled (i.e., outliers), only for revealing certain image details and overall spectral response. When undergoing spectral analysis (e.g., statistical, classification), these should remain unmodified in all cases, except for optical effects normalization which is the same process for each image. Although it is adequate to bring such corrections when conducted appropriately, in an

ideal setting, none would be required, but this is untrue for most radiometric sensors or remote sensing instruments.

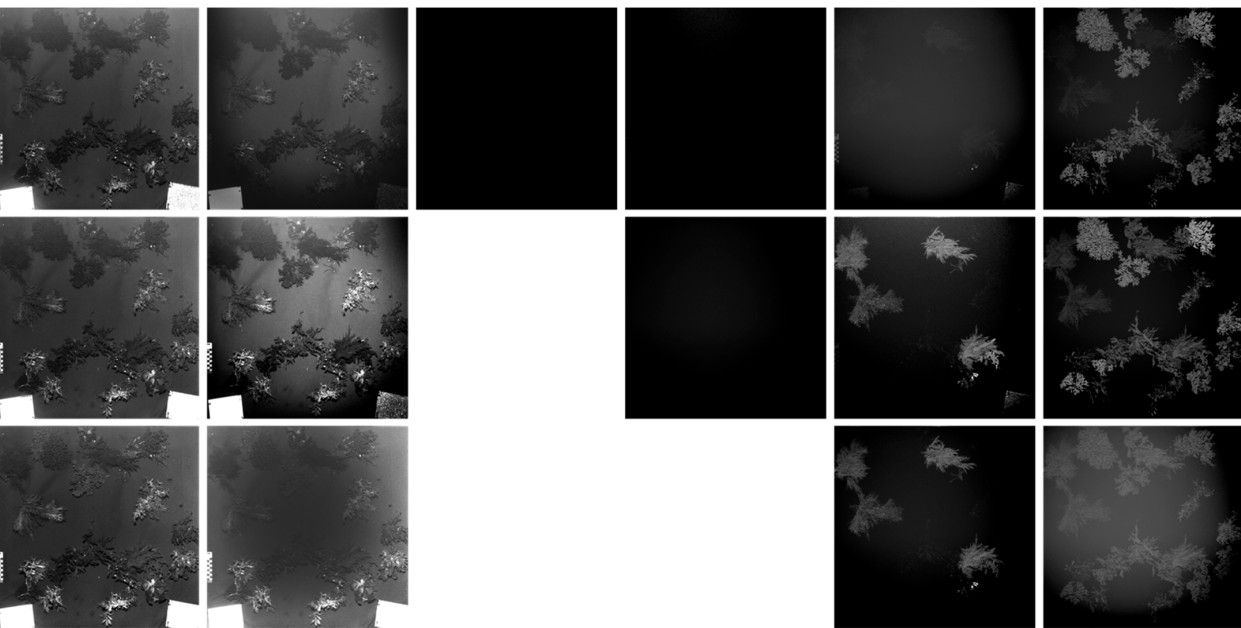

**Figure 5.** Images recorded through spectral response of macroalgae and species from excitation at 450 nm (top row) from left, reflectance at 450 nm without filter, 450 nm with bandpass filter, emission at 490 nm, 520 nm, 580 nm, and 685 nm; excitation at 490 nm (middle row) from left, reflectance at 490 nm without filter, 490 nm with bandpass filter, emission at 520 nm, 580 nm, and 685 nm, and excitation at 520 nm (bottom row) from left, reflectance at 520 nm without filter, 520 nm with bandpass filter, and emission at 580 nm and 685 nm.

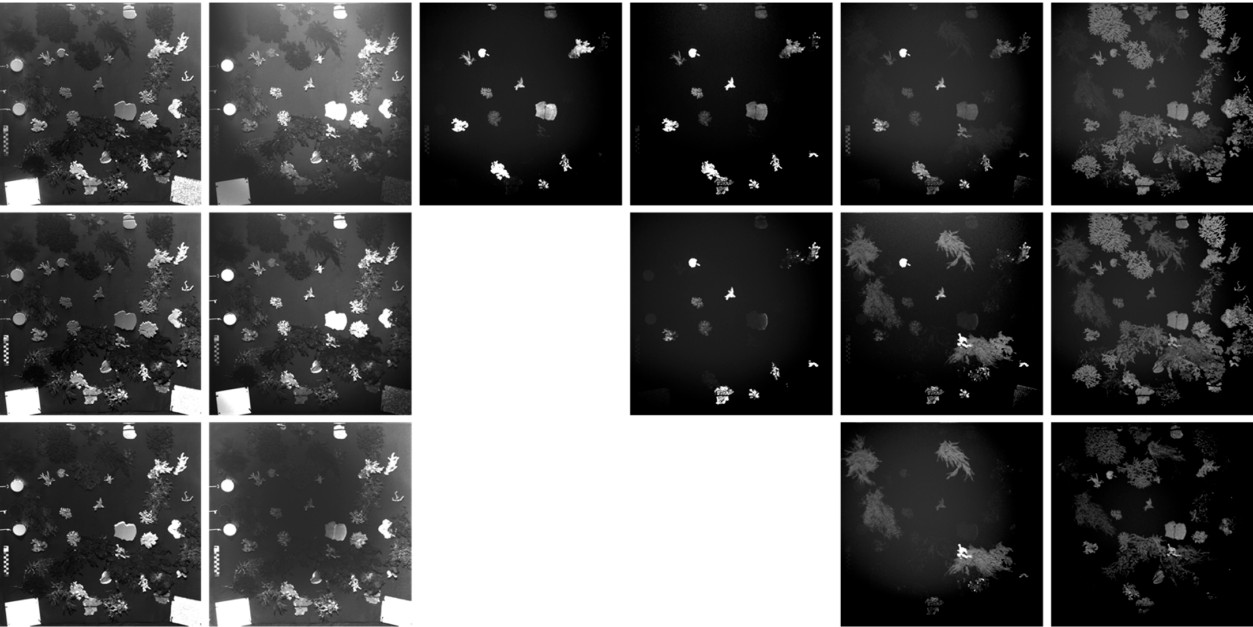

**Figure 6.** Images recorded through spectral response of macroalgae and coral species from excitation at 450 nm (top row), and from left, reflectance at 450 nm without filter, 450 nm with bandpass filter, emission at 490 nm, 520 nm, 580 nm, and 685 nm; excitation from left, at 490 nm (middle row), and from left, reflectance at 490 nm without filter, 490 nm with bandpass filter, emission at 520 nm, 580 nm, and 685 nm, and excitation from left at 520 nm (bottom row), and from left, reflectance at 520 nm without filter, 520 nm with bandpass filter, and emission at 520 nm, 580 nm, and 685 nm.

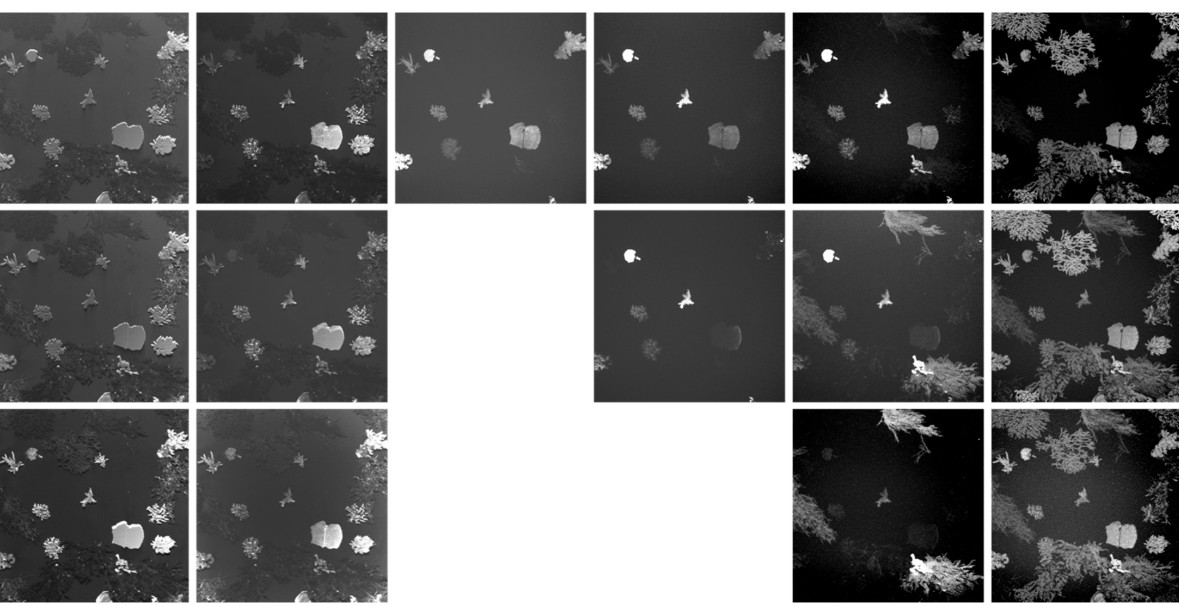

**Figure 7.** Images recorded through spectral response of macroalgae and coral species, from excitation at 450 nm (top row), from left with reflectance at 450 nm without filter, reflectance with bandpass filter, and emission at 490, 520, 580, and 685 nm; excitation at 490 nm (middle row), and from left with reflectance at 490 nm without filter, reflectance with bandpass filter, and emission at 520, 580, and 685 nm, and excitation at 520 nm (bottom row), and from left with reflectance at 520 nm without filter, reflectance with bandpass filter, and emission at 580 and 685 nm.

### 3.1.1. Macroalgae Fluorescence Imaging Scenario

Fluorescence imaging results were in accordance with certain known fluorescence characteristics in macroalgae of the three color classes [30]. Red macroalgae can be singled out by their response at 580 nm emission (a wavelength within a certain fluorescence response range), especially from 490 nm excitation, but also at 520 nm (Figure 5). Fluorescence response was not as intense from 450 nm excitation. Comparatively, green macroalgae showed the most fluorescence response at 685 nm, from excitation at 490 nm, compared to brown macroalgae which were also expected to emit fluorescence in this wavelength range, both resulting in chlorophyll-a fluorescence emission. This fluorescence intensity in green and brown macroalgal response is somewhat mirrored at 450 nm excitation but overall, less intense than after excitation at 490 nm. However, the fluorescence intensity response difference seemed lessened between brown and green macroalgae at 520 nm. On another note, overall fluorescence in red macroalgae at 580 nm appears fainter in images than that of other algal color types at 685 nm, and was more difficult to threshold from background illumination and highlight in post processing (i.e., for visual purposes). This may simply suggest a lessened physiological response in these red specimens ([30], where more laser source power (if the algae are not overwhelmed by the increased intensity) or receiver gain is needed while keeping the same imaging parameters. It is also possible that we are not observing at the optimal emission wavelength, although we have observed the range to quite broad from excitation from 450 nm to 550 nm in other red macroalgae [30]. Other emitter wavelengths could potentially generate more fluorescence but must remain in the range of laser diodes manufactured. Filter transmission could also affect observed signal intensity.

### 3.1.2. Macroalgae + Coral Fluorescence Imaging Scenario

In a mixed macroalgae + coral scenario (Figure 6), macroalgae showed the same fluorescence patterns as when imaged without coral. Macroalgae placement in this 2nd scenario is slightly different from the 1st since some specimens partially detached over the course of 1–2 days in the slow-moving current and were (or not) replaced by similar specimens. Coral did show typical fluorescence response as expected in the literature

through the 490 nm, 520 nm, 580 nm, and 685 nm fluorescence channels from excitation in the 450 to 520 nm range (Figure 6). While these will not be discussed in detail, certain species show drastic difference in fluorescence response from others. For example, a *Montipora capricornis* specimen in the image center, at excitation 450 nm excitation (top row), strongly fluoresces at 490 nm, and decreasingly so towards 580 nm. Comparatively, a *Montipora digitata* specimen, slightly right from center, shows a very strong fluorescence response near 580 nm from excitation at 490 nm, more so than at shorter emission wavelengths. The specimen also fluoresces considerably more than other imaged species (except *Psammocora stellata*, located halfway from center to top left image corner). The soft-bodied corals *Nephthea* sp. and *Xenia umbellata*, located approximately half-way and three-quarter ways from center, respectively, appear to emit fluorescence near 685 nm only, and the fluorescence response does not seem much influenced by excitation wavelength. While some species or genus may show undifferentiable spectral response, these spectral characteristics in coral as well as macroalgae are the basis for building an automated spectral response classifier.

### 3.2. Illumination Falloff Correction

To counteract this vignetting of mechanical (i.e., partial obstruction) but also of pixel (i.e., dependent of incident angle of light onto the light sensor) nature [31], observed in all images produced so far, we provide images for which the exterior border was cropped uniformly (Figure 7), effectively resizing images from $1000 \times 1000$ to $650 \times 650$ pixels. The optical effect is hence noticeably reduced in these cropped images, allowing for more appropriate pixel intensity-based spectral response classification. A similar method might be used in a situation where images acquired by an optical sensor through a special lens (e.g., camera, scanning LiDAR) would be cropped during the process of writing the image/data to file.

### 3.3. Fluorescence Intensity Thresholding for Pixel Segmentation

With the objective of processing the spectral response datasets automatically during or following acquisition while performing an AUV survey, fluorescent substrate pixels of macroalgae and corals needed to be identified by the signal intensity, per specific spectral response channel. To accomplish this programmatically, a threshold value was used to identify and apply a binary mask (Figure 8a) to the original images, keeping the spectral response signal intensities intact for analysis and providing background-subtracted images (Figure 8b). By this method, it was possible to efficiently identify fluorescent vs. background non-fluorescing pixels and drastically reduce dataset size. This is essential for simplifying and perhaps having the ability to tailor subsequent substrate-specific spectral response analyses by accessing only required parts of the dataset (e.g., macroalgae only).

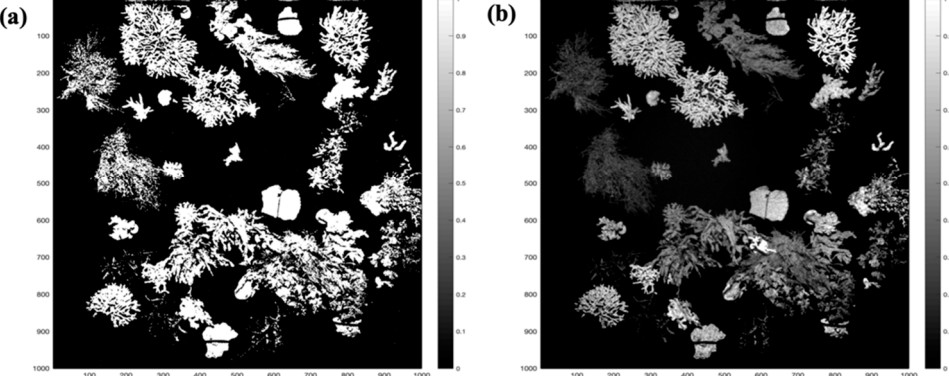

**Figure 8.** Fluorescent substrate pixel isolation and extraction from background non-fluorescent pixels via thresholding: (**a**) Left: A composite of fluorescence at 685 nm is used here to create a binary mask to select green + brown macroalgae, 580 nm for red macroalgae, and 520 nm for coral (but other wavelengths also can identify "coral" pixels); (**b**) Right: Fluorescent pixel image obtained from extracting 685 nm fluorescent pixels from non-fluorescing background pixels via thresholding.

### 3.4. Contrast-Related Image Quality Metrics

### 3.4.1. Resolution

The smallest and most accurately measurable and resolvable feature is in the fluorescence scale on the elastic response images (Figure 9). Since the real-life scanned dimensions are approximately 900 × 900 mm, and the black and yellow alternating lines near the ruler upper extremity are 1 mm thick each, resolution is therefore in the vicinity of 0.9 mm in the elastic images. Since it is a fluorescent scale, and that the inelastic fluorescence process is generally more diffuse (i.e., Lambertian) than an elastic, resolution is generally reduced in the inelastic images albeit appears to remain within the 1.0–1.5 mm range.

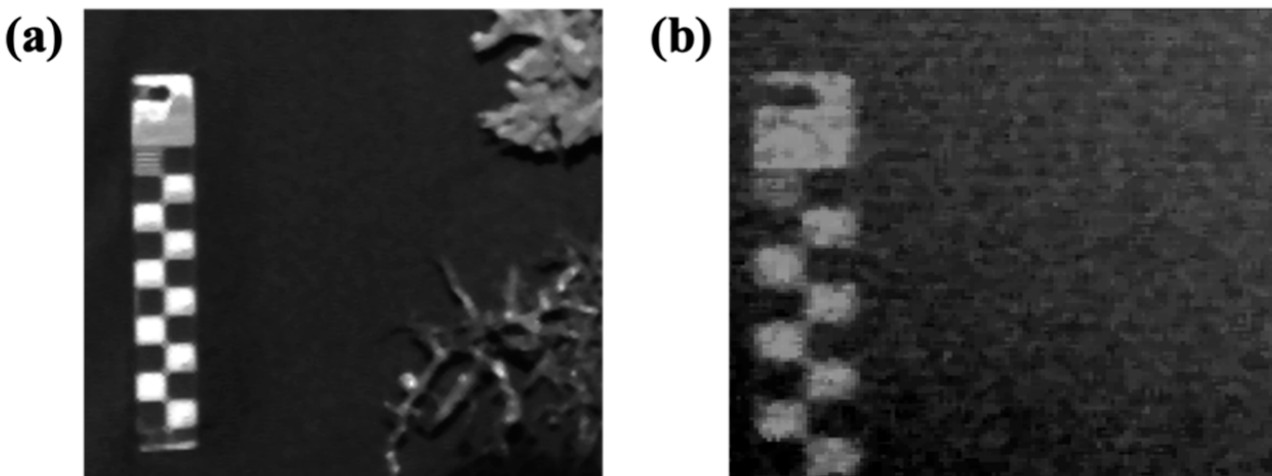

**Figure 9.** Fluorescent scale for resolution verification. White and black squares represent 10 mm segments. In the left image (**a**), the white and black lines visible in the upper left square represent 1.0 mm markings from 490 excitation and 490 emission (i.e., elastic), and in the right image (**b**), fluorescence at 580 emission from excitation at 490 nm.

### 3.4.2. Contrast Ratio (CR) and Contrast Signal-to-Noise-Ratio (CSNR)

Contrast Ratio and Contrast Signal-to-Noise-Ratio were evaluated for calibrated reflectance standards, green, brown, and red macroalgae, as well as for various coral specimens (see pixel subsample locations, Figure 10). A subset of 10 × 10 pixels from individual algal specimens of the three different color classes, as well as coral specimens, were used to calculate a series of 2 pi steradian average quantum yield of fluorescence values. Subset locations were selected on Spectralon™ reflectance standard targets, and where only coral (irrespective of species) or only one color class of macroalgae were present. A wide range of CSNR and CR values (i.e., quality = better contrast and higher SNR) in reflectance and fluorescence responses is seen in spectral response datasets (Figure 11). CSNR and CR for Spectralon are only shown in elastic response as any signal at other emission wavelengths would be considered as filter leakage (however, see Figure 6, 490520), eventually correctable by notch filter and better bandpass filter transmission cutoff values. Since reflectance targets are so much more reflective than nearby fluorescence-capable biological substrates and fluorescence is a much weaker signal, fluorescence-based spectral response analysis is unlikely to be "tainted" by the elastic/reflectance component. Additionally, CSNR and CR in elastic images are lower than other emission wavelengths for macroalgae, in comparison to coral.

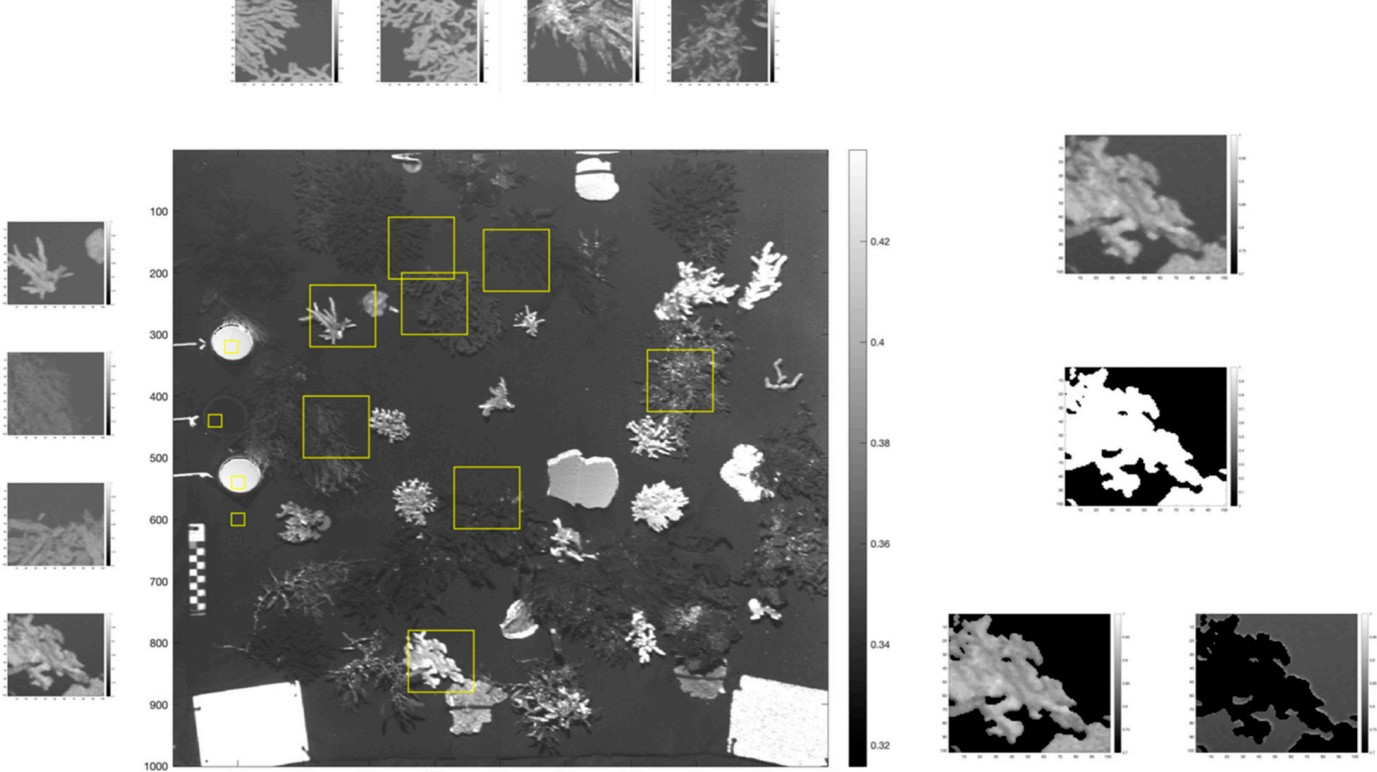

**Figure 10.** Contrast Ratio (CR) and Contrast Signal-to-Noise-Ratio (CSNR) macroalgae and coral subsample locations (large yellow squares), as well as reflectance standard subsets (small yellow squares. Macroalgae and coral specimen subsets are shown vertically to the left and top of the main center image. To the right of the main image, coral subsets (top right) are shown, with binary selection filter (from image pixel fluorescence threshold value) (middle right), and programmatically (binary fluorescence filter) selected fluorescent (to right of main image, left subset) and non-fluorescent background (to the right of main image, right subset).

By using fluorescence intensity thresholding to identify algal pixels, "fluorescence-capable" pixels are correctly identified, however, they score low in CSNR and CR index ratios because of a similarly reflective surrounding background. Further, using fluorescence emission at 580 nm for identifying red macroalgae rather than at 685 nm did not show substantial change or improvement in the contrast-related index ratios, but would be a useful approach in generating "fluorescence-capable pixels" map, as it would be for other potential emission wavelengths for other substrates of interest.

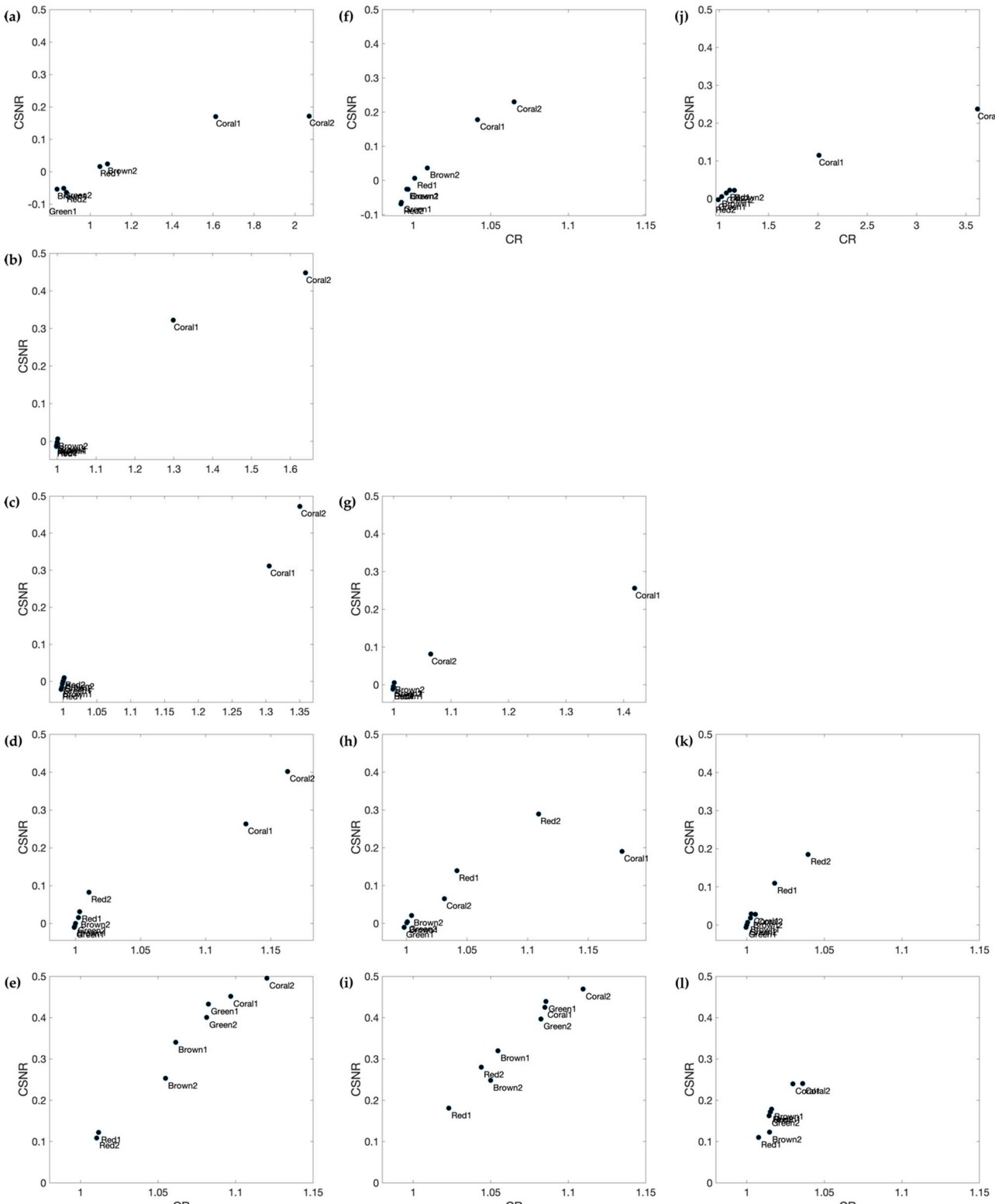

**Figure 11.** Contrast Signal-to-Noise-Ratio (CSNR) and Contrast-Ratio (CR) indices calculated on select spectral response image subsets of macroalgae and coral specimens, from excitation at 450 nm (left column), from top to bottom with (**a**) reflectance at 450 nm with bandpass filter, and emission at (**b**) 490, (**c**) 520, (**d**) 580, and (**e**) 685 nm; excitation at 490 nm (middle column), from at top to bottom with (**f**) reflectance at 490 nm with bandpass filter, and emission at (**g**) 520, (**h**) 580, and (**i**) 685 nm, and excitation at 520 nm (right column), from top to bottom with (**j**) reflectance at 520 nm with bandpass filter, and emission at (**k**) 580 and (**l**) 685 nm.

### 3.5. Fluorescence-Related Imaging System Variance Measure

Opto-mechanical instrument and biological-related image acquisition stability shows low overall variance between replicated images. The results, however, show several locations with higher replicate-wise pixel intensity variability. In several instances, gammarids were observed moving in the water column and nearby some algal specimens. These organisms moving within the algae, the latter which emit fluorescence, during imaging may be the reason higher values are observed at x–y locations 650–500 and 850–350. However, the main observable feature is represented by 2 soft coral species capable of movement within the timeframes between replicate images (x–y: 700 to 950–500 to 700). Moreover, fluorescence variability does not appear to be any more important than in the background when looking at the un-denoised image (Figure 12a).

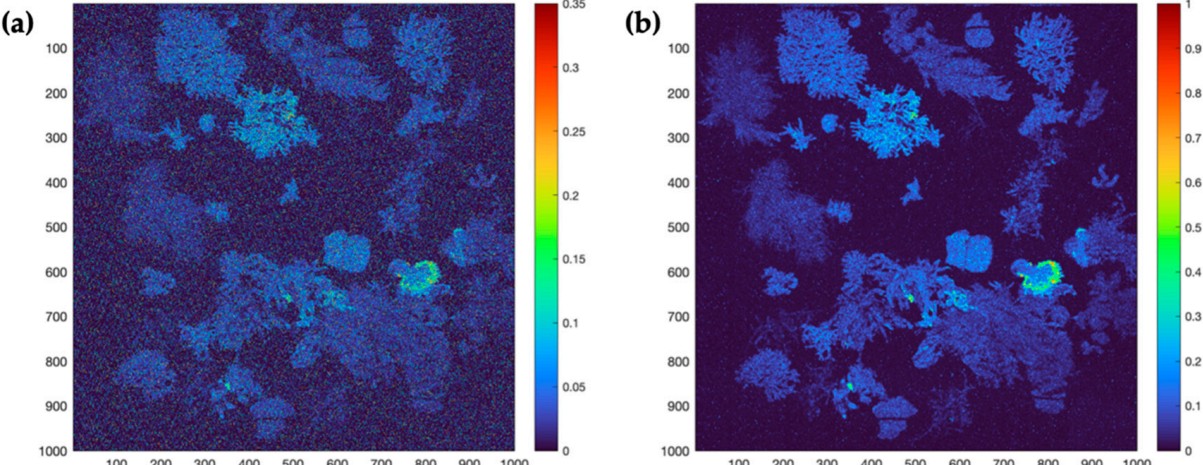

**Figure 12.** Pixel intensity variance between image replicates (**a**) not denoised; (**b**) denoised using 3 × 3 median filter) generated by laser excitation at 490 nm, and corresponding fluorescence at 685 nm in various macroalgae and coral species. Note an area of high variance that can be attributed to soft coral-associated structure and polyp contraction between image takes (i.e., approx. 30 min).

### 3.6. Irradiance on the Pixel-Photon Model for CW Line Scan

Laser diode beam spot sizes were measured on target surface and were 1.5 ± 0.1 mm for 450 nm and 490 nm laser diodes, whereas the 520 nm laser diode spot size on target measures 2.0 ± 0.1 mm. Since laser scan speed was set to 20,000 mm/s, effective dwell times for 450 nm and 490 nm laser diodes on target were 75 µs, whereas dwell time for the 520 nm laser diode was more in the range of 75 µs (Supplementary Materials Table S2). Using water IPO values from [32], estimate values were calculated for the wavelengths under study (Supplementary Materials Table S3), and used to provide an estimate calculation of photons per unit area (Supplementary Materials Table S4).

### 3.7. Reflectance and Practical Fluorescence Efficiency

Pixel reflectance values near 3% are also in agreement with reflectance measurements for macroalgae in other studies [33–35]. Practical fluorescence efficiency in chlorophyll-a present macroalgae fell within or near the few available published literature estimates [36], as did values for coral (i.e., 4–6%) [37] (Table 1). Some variability is expected as the substrates are of biological nature. Since the PMT QE at elastic wavelength of 490 nm (11%) is slightly higher than inelastic fluorescence response at 685 nm (approx. 10.2%), presented values are adjusted accordingly by correcting raw image pixel intensity values before practical fluorescence yield calculation.

**Table 1.** Substrate reflectance and practical fluorescence yield values, corrected for water column attenuation effects.

| Substrate Type | Pixel Reflectance | Practical Fluorescence Efficiency |
|:---:|:---:|:---:|
| Green macroalgae | 0.0295 | 0.0219 |
| Brown macroalgae | 0.0287 | 0.0161 |
| Red macroalgae | 0.0302 | 0.0149 |
| Coral-hard | 0.0967 | 0.0260 |

## 4. Discussion

Emphasizing on laser serial imaging and detection of macroalgae and coral, applied to large km-scale underwater biological surveys, our work expands on previous trials for underwater imaging of living benthic substrates such as coral [13–18,23]. We aim to further improve underwater remote sensing imaging means, with the objective of identification and classification of benthic flora and sessile fauna. Our study also bridges the gap between laboratory [22,38] and more in situ experimentation in remote sensing detection and classification of macroalgae, by working in the laboratory with yet realistic water column volume and optical characteristics as well as AUV working distance conditions. Using laser serial imaging techniques to generate and record spectral response in macroalgae as well as coral, we demonstrate their potential for detection, quantification, and differentiability in the context of future large-scale coastal ecology-driven benthic macroalgal and coral surveys.

### 4.1. Generating Spectral Response with the Proposed Imager Design

While being able to provide high resolution benthic substrate images at an adaptable distance of at least 2.3 m, the underwater multispectral laser serial imaging system we designed is efficient in highlighting some of the main fluorescence response signatures representative of macroalgae and coral. In algae, chlorophyll-a dominant green macroalgae were identifiable as predicted via their strong fluorescence response at 685 nm. This was also observed in brown macroalgae but to a lesser extent than the green specimens. This fluorescence response appears to be excitation wavelength-dependent, where "bluer" excitation at 450 nm and especially 490 nm was more efficient in generating fluorescence in green macroalgae, whereas "greener" excitation wavelength appears to have narrowed the gap in visible fluorescence response intensity between green and brown macroalgae. If this trend continues further towards the green wavelengths, brown macroalgae could be somewhat more differentiable by their higher fluorescence from excitation into the "far-green" wavelengths (i.e., 532 nm), as supported by other works demonstrating fluorescence in various macroalgae of the three color types [39]. Red macroalgae may be the easiest to differentiate from other colored species by their fluorescence response in the 580 nm range, where green and brown macroalgae color types do not show easily observable fluorescence in this wavelength range. Fluorescence in the red macroalgae was somewhat less intense comparatively to other color groups at 685 nm, and this may be due in part by the structural nature of the selected algae. Selected and available reds were effectively thinner than greens and browns, suggesting that thicker or more dense photosynthetic tissue could generate more fluorescence for the same excitation energy given on its surface. Red macroalgae are, however, known to vary in thickness between species, so this should not be considered a trend in red macroalgae per se, but possible in macroalgae of different thicknesses.

In coral, high-resolution imagery is also possible from acquired data, and as for macroalgae, quite dependent on laser beam diameter or spot size on the target surface. For this study, all laser wavelengths and combined emission filters provided quality high-resolution spectral response images for eventual classification. The 520 nm wavelength emission filter was also important in the sense of allowing rapid differentiation between coral and macroalgae, as the latter do not or emit very little at this wavelength. This allows for potential rapid "first-step" segmentation or classification between these two substrates occurring in a mixed coverage setting. Differentiation between genus, species,

and structure (i.e., flat/erect, soft/hard) in coral is not easily visible in images at first, but possibly for some coral specimens (i.e., soft coral species) which emit more specifically at 685 nm (or perhaps our imager, in its current configuration, only allows us to see limited emission wavelength ranges), which is a response solely due to their symbiotic microalgae. Machine learning-based spectral classification methods should provide better means for differentiating between algal color types and coral genus/species, considering that multiple spectral response parameters can be evaluated simultaneously.

Initially, use of a 405 nm laser diode in series with the other laser sources was intended for extended classification excitation range on coral (e.g., [24,40]), but unfortunate difficulties (i.e., alignment issues, beam diameter properties, galvanometric mirror surface coatings) during the tests prevented us from generating spectral response images for which optical effects could be corrected for. Macroalgae would have also possibly benefited from this additional excitation wavelength based on an absorbance curve shown in literature [41], and from the images that were obtained during imaging trials (see Appendix A Figure A1). Macroalgae spectral response would also have possibly benefited from a 505 nm excitation source to provide further possibility of discrimination between brown and green fluorescence response, but observation via the 520 nm imaging source suggests this may not have provided as much benefit as from excitation at longer wavelengths, such as 532 nm. The 505 nm wavelength is sitting close to the 490 nm and the 520 nm, but the fluorescence response increase appears gradual from 490 nm to 520 nm. Literature also suggests relatively low differentiability in macroalgae by independent reflectance measurements within this range [33–35,41], but could be of use in differential reflectance measurements [34].

A potential additional value to underwater remote sensing biological surveys is the addition of practical fluorescence yield as a proxy measure of macroalgal or coral health, somewhat comparable to phytoplankton variable fluorescence measurements [42]. By itself, the elastic signal acquired during imaging can provide high resolution images of these biological substrates, but do not readily give their health status. Additionally, inelastic fluorescence response can provide an indication of the nature of the imaged substrate, for example, detecting fluorescent living substrates, algal or coral-like, but with less resolution. A challenge thereby is in using both signal types to extract more information from the image data. For this, an understanding of water column IOPs in a given survey location, coupled with a thorough elastic and inelastic sensor calibration (i.e., image reflectance and fluorescence intensity value "behavior" in different imaging situations) are required. In situations where fluorescence is detected and imaged, but differences in intensity could be associated with the substrate's structure (e.g., kelp blade thickness) [30,39,43], fluorescence yield in a healthy specimen should remain the same for a given species regardless of structural variation. Importantly, measurement conditions should therefore be considered as important for this type of evaluation.

### 4.2. Creating Radiometrically Correct Images for Spectral Analysis

While the objectives of our study were to evaluate multispectral imaging and detection methods for the generation of spectral response data sets, the receiver optics we used to record reflected light and emitted fluorescence were the source of additional complications in the normalization of images for possible intensity-response use and statistical/classification analysis. In hindsight, additional steps could have been taken in planning for the normalization procedures by background correction measures and the effect that the filters may have in creating a drop-off in illumination. Additionally, in preparing a more definite and "ready for use" version of such an imager, one would benefit from a more thorough ray-tracing analysis to efficiently choose the optical receiver components. Depending on intended field of view (e.g., 60 degrees (−30 to +30 centered on nadir), 90 degrees (−45 to + 45 centered on nadir), etc.) and therefore scan area width at expected working distances underwater, collecting optics should be designed to receive as much light and signal as possible while maintaining distortion effects to a minimum. These

steps would in turn facilitate programmatic image correction and possibly provide better results (e.g., [29]). In our present situation, in cropping images to an area size where the illumination drop-off effect was sufficiently corrected for, we believe we have been able to generate spectral response datasets adequate for fluorescence intensity response analysis. In a subsequent version of such an imager, each laser source and emission filter pair would require individual radiometric calibration to ensure optimally acquired images and/or remote sensing spectral response data (e.g., [44]). Further, alternatives in filter dimensions and characteristics, as well as a receiver-type front lens could alleviate and even eliminate some of these optical effects. Optimizing filter optics, such as improving transmissivity, and the addition of narrow passband notch filters to block elastic response leakage more efficiently, could substantially improve fluorescence detection.

Difficulties in generating quality images may arise when surfaces or substrates of different reflective index and/or fluorescent response are within the same image (e.g., Figure 10). Satellite imagery (e.g., Landsat) pixel saturation is, for example, frequent in situations where forest fires are occurring [45], and useful in their detection. In our situation, it was difficult to assess exactly how the detector would respond to these substrates in terms of signal saturation. This situation resulted in some of our mixed substrate benthic scene recreations to be more saturated on certain corals, reducing detail in areas of these reflectance images while attempting to maximize fluorescence detection with increased signal gain. By increasing laser power output, one could expect a certain increase in fluorescence response. However, this also causes pixel saturation from highly reflective surfaces. We show that highly detailed images can be made, at the cost of losing some contrast or intensity in one end of the surface reflectivity/fluorescence spectrum or the other (i.e., algae/coral vs. artificial surface). This could be remediated by optimizing each optical receiver channel to the type of targeted substrate to characterize in a specific situation. For example, a specific sensor could be reserved for highly reflective surface characterization, by having a neutral density filter in place (i.e., verified to keep dynamic range and not saturate the sensor) or more appropriate gain setting, while another sensor recording reflectance at the same wavelength in a natural/biological substrate, or fluorescence at another channel would not have the same neutral density filter to attenuate this light, and/or not have the same gain/amplification settings if necessary.

### 4.3. Consideration for Wavelength-Dependent in-Water Differential Refraction Effects

With a multispectral imaging instrument comprised of multiple laser excitation sources, great care must be taken to precisely align beams so they may follow the same path to the target at intended operating distances. Additionally, a greater path length from instrument to target increases this requirement for precise alignment. Failure to do this can result in a slightly different start location at the beginning of a scan. This could also be more noticeable as the scanning area is increased and more visible at the edges of the scanning field. Higher resolution systems may also suffer more significantly from this problem as the laser spot size is smaller and gives less opportunity for adjacent pixels to partially overlap. In hindsight, our setup showed good alignment between laser sources for the distance at which images were taken. Visualization of the spectral response layers' pixel alignment via a GIS software, used for preliminarily pixel segmentation analysis, showed relatively low offset between spectral response layer pixels, except nearing image borders. This effect can easily be minimized by precise alignment procedures when working on a final version of the imager that has undergone optimal component selection through imaging trials. Laser wavelength color deviation (i.e., refractive index-dependent) and absorption can also be accounted for in later developmental stages of the imager. For example, imaging a dot-matrix technical target at desired wavelengths, which allows calculation for beam displacement in relation to evenly spaced markers in both x and y dimensions.

### 5. Conclusions

In creating a multispectral imager for an underwater AUV, optical and electronic component choices must be made to coincide with the deployment platform operating characteristics and requirements. Laser sources are an appropriate choice in preparing an instrument for imaging in locations where scattering may be a restricting factor compared to broader or less focused light sources (e.g., by flash lamp). Current laser technology limits useable imaging wavelengths for macroalgae and coral to somewhere between 355 and 532 nm, but many CW laser diode options are available from 405 nm, 445 nm, 450 nm, 465 nm, 488 nm, 505 nm, 510 nm, 520 nm, and 532 nm (albeit more wavelengths are becoming available in the 540–550 nm range). Comparatively, stable, powerful, and high repetition-rate pulsed lasers are available in 355 nm, 473 nm, and 532 nm wavelengths. In its current form, the emitter system remains simple in its design by having a low number of excitation sources, which may be a challenge but still feasible to implement (i.e., interweaving sequential timing of specific laser wavelength scanning and acquisition process). This relatively unexplored approach could characterize spectral response in the PAR (350–800 nm range via multiple near simultaneous multi-wavelength excitation–emission processes in substrates of interest.

Our work shows the potential for detection and classification in biological benthic substrates such as macroalgae and coral by their reflectance and fluorescence response. The ideal working distance-to-target is in the range of 1 to 5 m for an optimal detection of chlorophyll-a fluorescence, albeit this can possibly be extended to 10 m in clear waters [46]. Additionally, the observable signal range is dependent on the intensity of fluorescence emitted, which depends on the laser source intensity and power density. This detection limitation is mainly due to the higher absorption coefficient of seawater for light at 685 nm (i.e., Chl-a fluorescence peak). Light absorption and scattering reference values for the wavelengths used in this study suggest an attenuation of the fluorescence of nearly 0.5 m$^{-1}$, or loss of 50% of the signal in only 1 m of pure seawater. This absorption (or attenuation) is, however, much less at lower wavelengths, from 400 to 600 nm, 0.01 to 0.244 m$^{-1}$, respectively, which facilitates light 'delivery' from source to imaging target, as well as fluorescence and overall light detection in this lower range in which coral and macroalgae also show spectral response. In the context of future environmental coastal surveys, where optical condition may vary locally, it is best to optimize signal-to-noise ratio of fluorescence via using the most powerful but practical laser sources possible, and high transmission optical components, while adjusting parameters during the survey (e.g., detector gain, laser intensity). However, this imaging method is still subject to water optical quality, as all other imaging methods, and surveys should be planned to fit with local water optical conditions to optimize dataset and image quality. AUV-based surveys can reduce many of these inconveniences by working as close as possible to the imaging substrate, and wave/wind conditions that AUV vehicles which are already built to compensate for can be minimized.

A high-resolution multispectral receiver could be an integral part of such a multi excitation wavelength system by having many (i.e., 16, 32, etc.) identical sensors recording in parallel at different narrow wavelength bandwidths. This receiver would bring the possibility of generating a more accurate and detailed spectral response signature for spectral discrimination in target surfaces. Additionally, pulsed lasers in the 355 nm, 473 nm, and 532 nm range for pulsed LLS/serial LiDAR would give the advantage of range resolution and 3D point cloud generation. Moreover, dual simultaneous (or near) pulsed laser sources, such as 532 nm and a 473 nm blue laser, or a 355 nm would allow for work on detection and spectral discrimination by fluorescence and differential reflectance, which has shown some promise in obtaining additional spectral response classification measures.

**Supplementary Materials:** The following supporting information can be downloaded at: https://www.mdpi.com/article/10.3390/rs14133105/s1.

**Author Contributions:** Conceptualization, M.H., E.R. and F.D.; Methodology, M.H. and F.D.; Data analysis, M.H.; Resources, P.A. and M.P.; Writing—original draft preparation, M.H.; Writing—review and editing, M.H., F.D., M.P. and P.A.; Supervision, E.R., F.D., M.P. and P.A. All authors have read and agreed to the published version of the manuscript.

**Funding:** This research was in part funded by Sentinel North to P.A. and M.P. and the Link Foundation, the latter providing a major portion of funding for the Harbor Branch Oceanographic Institute Summer Intern Program for which some aspects of this work were made possible. PhD Scholarship to M.H. was paid by Sentinel North and partly by Arcticnet.

**Data Availability Statement:** The data presented in this study are available on request from the corresponding author.

**Acknowledgments:** Many thanks to Harbor Branch Oceanographic Institute (HBOI) at Florida Atlantic University's Ocean Optics team for support (B. Ramos, C. Strait, A. Tonizzo), the Link Foundation at HBOI, and helpful administrative personnel for making the project possible through the summer internship program, and V. Sommers at ORA, Fort Pierce, FL.

**Conflicts of Interest:** The authors declare no conflict of interest.

**Appendix A**

**Table A1.** Macroalgal and coral genus, species, and group names used during imaging trials (distribution obtained from the World Register of Marine Species database [47].

| Type | Genus | Species | Color Class | | Known Distribution |
|------|-------|---------|-------------|--|--------------------|
| Macroalgae | *Codium* | sp. | Green | | Eastern Florida Coast/Atlantic/Caribbean |
| | *Sargassum* | sp. | Brown | | Eastern Florida Coast/Atlantic/Caribbean |
| | *Dictyota* | sp. | Brown | | Eastern Florida Coast/Atlantic/Caribbean |
| | *Padina* | sp. | Brown | | Eastern Florida Coast/Atlantic/Caribbean |
| | *Grateloupia* | sp. | Red | | Eastern Florida Coast/Atlantic/Caribbean |
| | *Halymenia* | sp. | Red | | Eastern Florida Coast/Atlantic/Caribbean |

| Type | Genus | Species | Structure | Shape | Known distribution |
|------|-------|---------|-----------|-------|--------------------|
| Coral | *Acropora* | *austera* | Hard | Erect | Indian Ocean/Pacific Ocean/Red Sea |
| | | *cyatherea* | Hard | Erect | Indian Ocean/Pacific Ocean/Red Sea |
| | | *nana* | Hard | Erect | Indian Ocean/Pacific Ocean/Red Sea |
| | | *nasuta* | Hard | Erect | Indian Ocean/Pacific Ocean/Red Sea |
| | | *nobilis* | Hard | Erect | Indian Ocean/Pacific Ocean/Red Sea |
| | | *valida* | Hard | Erect | Indian Ocean/Pacific Ocean/Red Sea |
| | *Echinopora* | *lamellosa* | Hard | Flat | Indian Ocean/Pacific Ocean/Red Sea |
| | *Montipora* | *capricornis* | Hard | Flat | Indian Ocean/Pacific Ocean/Red Sea |
| | | *confusa* | Hard | Erect | Indian Ocean/Pacific Ocean |
| | | *digitata* | Hard | Erect | Indian Ocean/Pacific Ocean/Red Sea |
| | | *spongodes* | Hard | Erect | Indian Ocean/Pacific Ocean/Red Sea |
| | *Nephthea* | sp | Soft | Erect | Indian Ocean/Pacific Ocean |
| | *Pavona* | *decussatus* | Hard | Erect | Indian Ocean/Pacific Ocean |
| | | *frondifera* | Hard | Erect | Indian Ocean/Pacific Ocean/Red Sea |
| | *Pinnigorgia* | *flava* | Soft | Erect | Indian Ocean/Pacific Ocean |
| | *Plexaura* | *flexuosa* | Soft | Erect | Gulf of Mexico-Caribbean |
| | *Pocilliopora* | *damicornis* | Hard | Erect | Indian Ocean/Pacific Ocean/Red Sea |
| | *Psammocora* | *stellata* | Hard | Erect | Indian Ocean/Pacific Ocean/Red Sea |
| | *Seriatopora* | *hystrix* | Hard | Erect | Indian Ocean/Pacific Ocean/Red Sea |
| | *Stylophora* | *pistillata* | Soft | Erect | Indian Ocean/Pacific Ocean/Red Sea |
| | *Xenia* | *umbellata* | Soft | Flat | Indian Ocean/Red Sea |

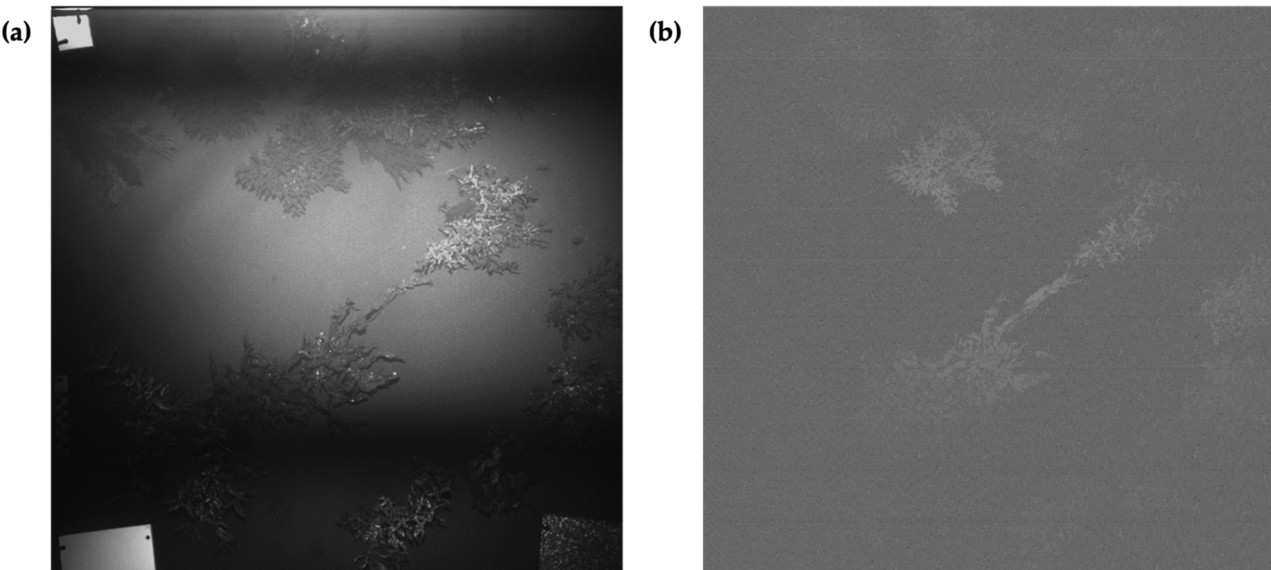

**Figure A1.** Laser-line-Scan images recorded through spectral response of macroalgae and species from excitation at 405 nm and (**a**) reflectance at 405 nm and (**b**) fluorescence emission at 685 nm on algal substrates only. Imaging parameters did not allow to use data provided by the 405 nm excitation source since radiometric correction would prove difficult at best.

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
