# Peer review of "Underwater Multispectral Laser Serial Imager for Spectral Differentiation of Macroalgal and Coral Substrates"

_remotesensing, doi:10.3390/rs14133105_

Round 1
Reviewer 1 Report
Comments to the Author
Review of the manuscript “Underwater multispectral laser serial imager for spectral discrimination of algal and coral substrates”
This study intends to improve current underwater imaging techniques for classifying coastal benthic flora and sessile fauna by underwater multispectral laser serial imaging. Multiple laser wavelength sources were used to scan and illuminate a recreated benthic environment (saltwater tank with 1.5 m W × 7.0 m L × 1.5 m H) with macroalgae and coral obtained from the low tide to shallow subtidal zone of the Florida Atlantic coast. This study demonstrated its potential for detecting and quantifying benthic macroalgal and coral surveys. The manuscript is well written with details, organized, and addresses a topic of great relevance to studies in shallow coastal underwater biological surveys. The data analysis methods are appropriate and support their conclusions. Therefore, this work deserves to be published in an important journal such as Remote Sensing.
However, the manuscript can still be improved. I highlighted some points that can make this work more adherent to the needs of marine researchers.
Important information is diluted throughout the text, such as: the limitations (depth, turbidity, weather conditions, etc.) to using this technique in a real environment, the logistics needed during fieldwork, and the cost of this equipment. I am sure that biologists, oceanographers, and geologists will be interested in this technique, so it is important to emphasize to these professionals when and where this technique can be used.
The choices of parameters to achieve success in this study were determined in a static and controlled environment. For instance, ”the choice of laser sources is based on the absorption and light emission curves of macroalgae and coral, as coincidentally as well as their transmission in water. The lasers were also chosen for the beam quality, allowing focusing for high resolution imaging in underwater conditions adjusted to a distance of 2.3m” (lines 176 – 180). Then, the parameters of the multispectral laser used on shipboard or autonomous underwater vehicles could be adjusted during the survey according to changes in depth, water transparency and target area? Does the action of currents, waves, and winds on shipboard or autonomous underwater vehicles affect the quality of the data obtained? May the natural environment light blend with the laser and interfere with the scan’s accuracy? Many questions about laser multispectral properties are discussed in great depth. I recommend bringing this discussion to the use of this technique in coastal areas involving all the adversities of this environment. Part of the Discussion could be used to clarify such issues.
The main conclusion (this technique may be used to detect and quantify benthic macroalgal and coral) and their limitations must be clearly presented in the Abstract and Conclusions.
The work has 30 pages distributed in the following sections: Introduction (2.5 pages), Materials and Methods (10.5 pages), Results (9 pages), Discussion (3 pages), Conclusions (0.5 page), and References (2 pages). I recommend condensing the Materials and Methods and/or sharing this content with the Supplementary Materials.
Considering these issues, I believe the results of this study could be used by other researchers in coastal environments and the manuscript will gain more interest from the Remote Sensing reader.
Reviewer 2 Report
Underwater multispectral laser serial imager for spectral discrimination of algal and coral substrates
by: Matthieu Huot, Fraser Dalgleish, Eric Rehm, Michel Piché and Philippe Archambault
The advancement of innovative underwater remote sensing detection and imaging methods, such as continuous wave laser line scan or pulsed laser imaging approaches can provide novel solutions for studying biological substrates and manmade objects/surfaces often encountered in underwater coastal environments.
The Authors expand current underwater remote sensing methods to combine macroalgal and coral surveys via the development of a multi- spectral laser serial imager design for classification via spectral response. By using multiple laser wavelength sources to scan and illuminate recreated benthic environments composed of macroalgae and coral, spectral responses can potentially be used to differentiate algal color groups and certain coral genus. Experimentally, three laser diodes (450nm, 490nm, 520nm) are sequentially used in conjunction with up to 5 emission filters (450nm, 490nm, 520nm, 580nm, 685nm) to acquire images generated by laser line scan pattern via high-speed galvanometric mirrors.
The structure of the article is considered and clear. The background and comprehensive review of the problem's literature were presented. Discussion and conclusions, on the basis of the research, are comprehensive and clear.
Specific suggestions are in attached pdf file, especially text and equations editing.

Reviewer 3 Report
Based on the development of a multispectral laser serial imager design, the authors did an indoor experiment to show the potential of spectral responses (the reflectance and the fluorescence emission respond to some positive laser beams) to classify algal color groups and certain coral genus. This is interesting and meaningful. Some suggestions are as follows:
- Line 176-178:The authors said that the choice of laser sources is based on the absorption and light emission curves of macroalgae and coral (i.e., in the visible spectrum), as coincidentally as well as their transmission in water. Can the absorption and light emission curves of macroalgae and coral be provided?
- Line 257: Is the corals live?
- Figure 2: I suggest the authors to provide higher-resolution images, and to mark the red (Rhodophyta), green (Chlorophyta), brown (Phaeophyta) algae, and the live corals, etc.
- If possible, I suggest to add some experiments in natural waters, and answer the question: whether the equipment and the method is feasible?
- If the effects of the classification are dependence on the incident angle and the observed distance?
Reviewer 4 Report
In general, I do believe this is a very interesting paper and up to the point it is well written.
When you lose the plot is the moment when you start mixing LIDAR results with images. And however sometimes LIDAR can be referred to as ‘imagining’ method it is not sctricly the truth.
The result of laser scanning is a point cloud and not an image in a classical sense, you can convert it to an image (geoitiff or just a projected mosaic) but in its core, it is a different type of data. Even if we talk about profile type laser units (in a classical sense but somehow similar to what you use) this still cannot be called an image.
The bottom line is in the paper you do need a proper distinction from 3D LIDAR data (that I think were part of an impute of the produced images – I cannot be 100% certain from the text) to the image in different bands that is a result.
It is partly delivered from the problem above but I have issues with seeing what was the aim of the study – geolocalisaion of the objects, 3D shape, just recognition?
This is a good technical paper but needs to be more ridable to everyone so is a reader knows everything about LIDAR and IR and nothing about multispectral he needs to understand the paper to and with mixing data with results a potential reader gets lost.
L240 why “?”
Round 2
Reviewer 3 Report
The questions have been answered in the responses, so I can suggest to accept the paper. However, the readers should pay more attention to whether the equipment and the method is feasible in natural waters, and how to eliminate the influences of the incident angle and the observed distance in practices. The authors need to discuss such problems in future research.